# Design, Manufacturing and Functions of Pore-Structured Materials: From Biomimetics to Artificial

**DOI:** 10.3390/biomimetics8020140

**Published:** 2023-03-29

**Authors:** Weiwei Chen, Lin Gan, Jin Huang

**Affiliations:** Chongqing Key Laboratory of Soft-Matter Material Chemistry and Function Manufacturing, State Key Laboratory of Silkworm Genome Biology, School of Chemistry and Chemical Engineering, Southwest University, Chongqing 400715, China

**Keywords:** biomimetic materials, porous structure, high performance, meta-structure

## Abstract

Porous structures with light weight and high mechanical performance exist widely in the tissues of animals and plants. Biomimetic materials with those porous structures have been well-developed, and their highly specific surfaces can be further used in functional integration. However, most porous structures in those tissues can hardly be entirely duplicated, and their complex structure-performance relationship may still be not fully understood. The key challenges in promoting the applications of biomimetic porous materials are to figure out the essential factors in hierarchical porous structures and to develop matched preparation methods to control those factors precisely. Hence, this article reviews the existing methods to prepare biomimetic porous structures. Then, the well-proved effects of micropores, mesopores, and macropores on their various properties are introduced, including mechanical, electric, magnetic, thermotics, acoustic, and chemical properties. The advantages and disadvantages of hierarchical porous structures and their preparation methods are deeply evaluated. Focusing on those disadvantages and aiming to improve the performance and functions, we summarize several modification strategies and discuss the possibility of replacing biomimetic porous structures with meta-structures.

## 1. Introduction

Nature has outstandingly constructed complex structural matter, providing structural design inspiration for humans to manufacture high-performance porous materials. As shown in Table 1, the mineralized layered structures in animals, plants, and tissues such as in bones [1,2], sponges [3], and diatoms [4,5,6,7,8], play a supporting role and have high compressive and elastic mechanical properties. The fibrous porous structures in grapefruit peel [9], wood [10,11], and bamboo [12,13] possess the characteristics of energy absorption, efficient material transport, and light weight. The fibrous porous structure can also be used to improve the functions of sound absorption, shock absorption, separation, and light weight.

Due to their high specific surface area, porous structures can further adsorb various functional elements in their cells to prepare multi-functional materials. For example, lightweight ultra-hydrophobic polyurethane bionic foams are manufactured, whose design is inspired by the self-cleaning multi-scale structure of lotus leaves and the porous structure of diatoms. The bionic foam is composed of micro- and nano-scale hydrophobic cells, realizing self-cleaning and efficient oil-water separation functions [14]. In addition, inspiration from the hierarchical ordered porous self-supporting structure of diatom cell membranes has been drawn for constructing highly ordered porous protein-chitin mineralized composites. Due to the self-supporting mineralized structure, the high Young’s modulus of the hybrid mineralized composite with the high loading property of SiO_2_ and TiO_2_ increases to 200% and 286%, respectively [6].

**Table 1 biomimetics-08-00140-t001:** Porous structure, cell morphology, specific surface area (SSA), features, and biomimetic applications of typical porous matters from nature.

Matters	Porous Structure	Cell Morphology	SSA (m^2^/g)	Features	Biomimetic Applications	References
bone	hierarchical mineralization	bleed hole	0.0394	high compressive strength	scaffolds for bone regeneration and bone replacement, and other tissue engineering materials	[1,2]
sponge	3D silicon network	bleed hole	30–400	elasticity, permeability, adsorption,	catalysts, energy separation	[3]
diatom	3D periodicity	honeycomb pore	9.7	high mechanical properties, light orientation	catalysts, separation, optics	[4,5,6,7,8]
pummelo peel	fiber layered porous	honeycomb pore	-	moisture retention, pressure resistance, cushioning, space limited region	energy absorption, functional integration, solar evaporation thermal materials, bionic foam composites	[9]
block wood	fiber layered porous	hollow	28.2	high transmission, high specific strength, energy absorption	intelligent electronics, EMI shielding, heat insulation, TiC/C ceramics	[10,11]
bamboo chip	fiber layered porous	hollow	0.0205	high transmission, strong toughness and strength, energy absorption	energy dissipation, light weight and vibration reduction laminated structural materials, renewable catalytic system	[12,13,15,16,17]

With regard to size, pores are usually divided into micropores (≤2 nm), mesopores (>2 nm and ≤50 nm), and macropores (>50 nm). Those pores probably show different adsorption behaviors, and thus exert different functionalities. As a result, multi-function design of porous materials usually needs hierarchical pores. The morphology and orientation of the pores also play important roles in functionalization. Preparing biomimetic porous materials is complex engineering, including figuring out the structure-performance relationship of natural matter, selecting optimized pore sizes and morphology, and combining and orienting different pores, as shown in Figure 1. For instance, amideoxime polymer membranes are designed with a complicated fractal structure from micron-sized macropore spans to nanopores, like the branch structure of vasculature. Since the macropore of fractal structure as the main channel accelerates water flowing and provides abundant adsorption sites, the uranium extract of the biomimetic membrane is 20 times higher than the membranes with only micropores [18].

Preparing methods for porous materials can be divided into two types: bottom-up physical assembly and top-down controlled foaming. However, the reasons why artificial porous materials may perform terribly, like insufficient strength and inhomogeneity, are due to the fact that traditional manufacturing technologies impede porous structural diversity and dimensional precision control. For instance, sol-gel and hydrothermal methods to prepare mesopores possibly bring structural disorder, poor dispersion, low crystallinity, and impurities [19,20,21]. The residual foaming agents from chemical foaming may exert an uncontrollable effect on the subsequent functional modification. Although etching methods can prepare pores with precise sizes, its high cost limits large-scale preparation.

Consequently, to make full use of biomimetic porous materials, we need more advanced methods to duplicate natural pores. It is also necessary to elaborate the influence of pore structure on the integration characteristics of mechanics and functional properties. This review summaries the widely accepted quantitative correlation between pore structures and material properties. This review also presents the theoretic basis for designing high-performance porous materials. Taking negative Poisson’s ratio meta-structured materials as an example, this review further prospects the development trend of porous superstructure-bionic design, and provides a new idea for manufacturing high-performance and multifunctional porous materials.

## 2. Design and Fabrication of the Bionic Porous Structure

### 2.1. Biomimetic Template Fabricated Pore Method

The template method is better characterized by all biomimetic hole preparation methods. The researchers generally utilize the template interface interaction to repeat the natural pore structure. Template methods can build pores with specific sizes from hundreds of micrometers to several nanometers. When we have learned the structure-performance relationship of specific pores from nature, we can use template technologies to manufacture those pores on a large scale. The degree of control over the template affects the effectiveness of biomimetic structure replication techniques, while artificial porosity has evolved from randomization to natural derivation to customization. Compared to traditional pore-forming methods, biomimetic template methods usually pay more attention to improving the performance of specific functions. This paper mainly introduces three pore control carriers, including ice template, naturally-derived template and customized-design template, and explores the technological change of pore structure imitating natural pore structure from randomness to controllability.

Ice template technology is widely used to prepare porous structure (5–50 μm) with regular geometric shapes [22]. The raw materials of the porous scaffold are first dispersed in water. Then, the dispersion is frozen to form ice crystals with regular shapes. The shape is adjusted by controlling the interaction between the crystal facets and the raw materials [23,24]. After removing the ordered array of ice crystal, the orientation of porous structures usually appears via freeze-dry technologies, and the bioactivity of low thermal stability polymer is reserved. For example, biomimetic oriented macropore of freeze-drying silk foams promotes moisture-absorbing and sweat-transfer performance from water-solubility of silk protein [25]. Then, the dendritical pores and pearl-like hierarchical pores of SiC scaffold can be prepared by ice templates for high mechanical performance [26]. Changing the cool rate and dispersion concentration affects the interface transition between SiC and ice crystal, causing a variation in the average wall thickness and pore morphology of the pearl cells. In addition to improving the mechanical performance, the porous structure can also be used to confer flexibility and tailor the mechanical performance. A polymeric porous foam, which was fabricated via freeze-drying, can mimic the native spongiosa layer of the aortic valve, in order to replicate the load-bearing role and tune the mechanical properties [27]. Such a conch-shell-like cross-porous structure via ice-templating provides permeability for epoxy resins, and energy dissipation in multiple crack deflection, for remarkable toughness and strength of conch-shell-inspired porous scaffold, like that of natural conch shell [28].

However, manufacturing porous materials by ice template technologies is facing challenges in scaling up. Moreover, its manufacturing efficiency and continuity is not high enough, and the pore size can hardly decrease to the submicron level. Bidirectional temperature control may be a promising solution. Such a strategy can efficiently and dynamically control the size growth of ice crystals, resulting in uniform porous materials, and improving the manufacturing continuity and efficiency [29].

Biological template methods, as a new kind of template method, can derivatively manufacture porous materials with finer and more complex structures, and integrate functions at the nano-micron level. Changing the size, morphology, and type of biological templates regulates the pore scale and pore morphology of biomimetic porous materials, and replicates their unique structures and properties. For example, using microorganisms as templates, the pore structure of porous materials is designed according to the biological template size (10 nm–100 μm), ranging from nanometers to microns. The researchers have taken rose petals [30], leaves [31], pine pollen [32], flax [33], bamboo [34,35,36,37], and other plant tissues as sacrificial bio-templates, and have synthesized oxide or carbon composites with hierarchical porous structure. The above bio-templates can be applied to synthesize biomimetic microporous materials in many aspects of drug release, dye degradation, electrode, antibacterial, sensing, and catalysis.

Notably, the bionic template method has the advantages of applicability, versatility, and multi-morphological controllability. The same kind of nanomaterials are introduced into different biological template types to prepare different micropore structures of biomimetic porous nanomaterials. Then, the researchers choose suitable templates for biomimetic preparation, according to personalized functions, to realize the high performance and multi-functionality of materials. For instance, using three kinds of branches; maple poplar branches, catalpa branches, and apricot branches; as biological templates to absorb the Zn^2+^ and Sn^2+^ ions resulted in pore sizes of ZnSnO_3_ ranging from 11~31 μm, 15~18 μm, and 2~6.3 μm, respectively. The formaldehyde gas sensitivities of the biomimetic porous ZnSnO_3_ composites are faster than those of manufactured fractional nanotubes or nanocages of ZnSnO_3_ [38].

According to the above examples, the naturally-derived template is directly from the bio-template. To effectively obtain a specific structure and integrated functions, imitating the key part of natural structures can effectively improve the customized characteristics of the material. Then, the synthetic biological components as biomimetic templates meet functional demands of porous materials to customize distinctive sizes and morphologies of nanopores (<10 nm). Biomimetic inorganic porous materials have the characteristics of self-assembly biological activity, chiral structural color, and selective permeability due to the replication of biological structures. Table 2 illustrates that the preparation of porous silica [39,40,41,42] and porous titanium dioxide [43,44,45,46] utilizes biological components including collagen and cellulose as inspirated templates. Additionally, biomimetic oxide can assemble in an orderly manner into chiral mesoporous structures via the interaction of bio-templates with surface functional groups such as carboxylic acid, hydroxyl group, and amino group based on the self-assembly of chiral biomolecules [47]. Typically, the chiral mesoporous structures of biomimetic inorganic nanomaterials have special functions of rainbow structural color and optical chiral response [48].

Many kinds of porous structures deriving from nature are prepared by the above template methods, while their high performances and multifunction are beyond natural structure. Biomimetic micropores own the characteristics of the templates for applying to photonic crystals, electrodes, electromagnetic shielding, photocatalysis, sensing, thermal management, medical treatment, and other fields. Nevertheless, the manufacturing and application of biomimetic materials face the following problems. First, the limitations of precise scale-controlled technology by a single template makes it difficult to combine complicated pore structures and functional integration. Secondly, the adaptability, stability, and continuity of the template method are unsatisfactory, because of the complexity of biological forms and the limited analysis of biological structures. Nowadays, the biomimetic template approach is in prospect of cooperating with new advanced technology strategies to expand the scope of pore scale manufacturing, and enrich the types of pore structures.

### 2.2. Biomimetic Porous Processing Technology

To release the limitations of controllable scale, hierarchy, and accuracy of pore structure in the process of the biomimetic template method, researchers have developed biomimetic porous processing technology to manufacture fine and super-large size porous structures. They simulated biological “top-down” structures by ordinary machining technologies for building integration architectures of biomimetic composite materials. For example, inspired by the grapefruit peel, the fractal pores of the carbonized porous material, including large pores (2 mm), macropores (40 μm) and nanopores (50 nm), are constructed by punching, processing, freeze-drying, and carbonization. The fractal pores of a solar biomimetic thermal material promote solar energy harvesting and photo-thermal conversion efficiency (Figure 2a) [9]. After carbonization, the bamboo chips’ reserved hierarchical porous structure is compounded with plastic particles by hot pressing technology to form bamboo charcoal plastic board. The hierarchical porous structures of bamboo charcoal plastic composite boards, similar to that of bamboo, including gaps, micropores, and voids, are derived from porous bamboo charcoal layers and rubber layers (Figure 2b) [31].

Next, seashells also are a type of biomaterial that shows great inspiration for biomimetics [49]. The “top-down” structures of porous materials can be produced by thermal treatment for conch shell and nacre [49,50,51,52]. There are other processing technologies to achieve “top-down” structures, such as electrospinning [53], phase-inversion [54], freeze-drying, [27] and supercritical technology [55]. For instance, after removing the oil phase, the hierarchically porous TiO_2_ nanofibers (≈300 nm) are fabricated via a microemulsion electrospinning. The hierarchical porous inner structures of TiO_2_ nanofibers, involving mesopores (22.7~26.2 nm) and macropores (≈57.4 nm), depend on the size of oil droplets [53]. As for the phase-inversion process, the proportion of evaporated solvent (tetrahydrofuran) is adjusted to customize the pore size of the polysulfone-block-polyethylene glycol membrane from 15.5 nm to 2.3 nm [54]. The tri-layer structures as “Film-Foam-Film” of biomimetic multilayer materials are manufactured by a combination of solution casting, electrospinning, and freeze-drying techniques. Therein, after removing the solvent, the porous structure of foam mimic layer is formed via freeze-drying [27]. Imitating the porous structure of natural straw, a three-dimensional lipophilic/hydrophilic layered structure can be constructed inside the hollow tubular structure via supercritical carbon dioxide. Additionally, the inner diameter size (0.4∼20.0 mm) and uniform foam pores of biomimetic tubular polypropylene foam are customized by adjusting temperature and pressure [55].

Furthermore, researchers have developed a series of “bottom-up” processing technologies, including layer by layer (LBL) self-assembly and 3D printing, for imitating natural complex porous structures and expanding porous structures ranging from macropore to micropore. Thereinto, the various functional structures are processed layer by layer into the whole layered structure of the microscale through LBL self-assembly technology to realize the multifunctional integration of bionic composite materials. For instance, based on the design inspiration of bamboo membrane structure, the hierarchical porous structure of porous membranes is fabricated by LBL assembly of graphene, Co_3_O_4_ nanosheets, and carbon nanotubes. The bamboo-like hierarchical opening pores with inner spacing (10–50 nm) and outer spacing (2.9 μm), enhance the capacity, cycle stability, and transfer efficiency of 2D–2D porous membranes [34].

Based on LBL self-assembly technology, 3D printing technology also can program and accurately design customized structures, finely stack macro-microscale structures, and automatically manufacture biomimetic composite materials with individual functions. The advantages of 3D printing technology include flexible and controllable processes, multi-dimensional fine structure reproducibility, and compatibility with rapid and large-scale manufacturing. For example, the complex biological porous structures involving cuttlefish bone (8–500 µm) [56], blood vessels (1.98–20 μm) [57], bones (~1 mm) [58], and grass stems (10 µm) [59] are accurately reproduced by 3D printing technology. Therein, 3D printing technology is widely applied to fabricate complex porous materials for additive manufacturing [60], medical treatment [61,62], energy storage [63], desalination [64] and other fields.

### 2.3. Biomimetic Mineralization Modified Pore Method

More than sixty kinds of natural mineralized porous substances exist, including oxides, sulfides, and weak salts, serving the roles of supporting, protecting, transporting, and promoting metabolism [65]. Organisms use the porous channels to build transport networks and to control nucleation and growth of active minerals precisely. Teeth and bones are formed in this way, which have excellent mechanical properties due to their stratified porous microstructure [66,67].

Bio-inspired by the stratified mineralization structure, researchers utilize porous scaffolds to mineralize functional nanoparticles and regulate their morphologies by adjusting the pore shape [68]. Changing the concentration of mineralized raw materials regulates grain growth and grain size, and then controls the pore size and porosity of the scaffold [69,70,71]. The mineralized activity of the carriers is an essential prerequisite for rich reactive groups and three-dimensional network pore structure [70,72]. The charged group, hydroxyl group, and the other reactive groups can provide some non-chemical bond action and intermolecular interactions for the mineralized carriers [72,73]. Thus, the adaptability and intermolecular interaction between mineralized carriers play important roles in controlling the particle size, crystal type, and morphology [68,74].

Meanwhile, the mineralization method is suitable for preparing tissue engineering materials due to the excellent biocompatibility of the natural porous scaffolds. Bone tissues have been used as porous scaffolds, providing the final biomimetic porous materials with excellent biocompatibility. In terms of tissue engineering, the porous bionic mineralized scaffold mainly applied to bone repairing and regenerating [69,70]. Next, the reduction of porosity and pore size can enhance mechanical strength, but it also needs to adapt to the internal environment of organisms and match biological activity. For example, the pore diameter of multilayer-modified PLGA (mPLGA) scaffold after the nano-HAPs mineralizing is reduced from 284.0 μm to 203.4 μm (Figure 3A) [74]. Similar to nano-HAPs, the biocompatible CaCO_3_ nanoparticles are occupied in an orderly way on the macro-porous network skeleton structure of the mineralized carriers, and the mineralized layer is generated to fill some defects of the pore scaffold (Figure 3B) [75]. The mineralized porous structure improves the compression strength of HAP-mPLGA scaffold by 10.78 times compared to the PLGA scaffold (Figure 3C), while accelerating bone regeneration [74].

The macropore structure is well-coincident with the demands of the size of transfer channels for tissue cell trafficking and activity substance transportation to promote bone healing and regenerating. Therein, the appropriate mineralized pore size may depend on the size of the mineralized particles and similar bone tissue pore sizes (50–450 μm). The range of mineralized pore size for the excellent biological activity is controversial at 80–250 μm [76], 300–500 μm [77,78] or 20–100 μm [69]. Some researches identify that the macropores at 20–100 μm are used for nutriment and oxygen transfer and adsorption to enhance biomineralized activity, and the others at 100–300 μm serve as rich active sites for bone growth, vascularization, and cell proliferation [69,78,79]. Especially, biomimetic mineralization adjusts the porosity of the scaffold to meet the requirements of different bone tissues [80]. For example, by regulating the calcium-phosphorus ratio close to bone composition, the porosity of composite scaffold was up to 70%, and its Young’s modulus increased from 1.19 MPa by 11.62 MPa; that resembles bone [81].

Furthermore, the enlightenment from the protection function of diatom shell, the biomimetic mineralization approach, is adopted to enhance the efficient utilization and stability of the pore structure of the functional carrier. The biomimetic mineralization method also is appropriate for other aspects, such as medical treatment, cell protection, and vaccine. First, a calcium phosphate shell was mineralized on the outer surface of porous ferritin to prepare a drug-targeted material for the application of medical treatment. For example, the mineralized shell structure has the function of pH stimulus responses and protective pore structures, which promotes the efficient utilization and safety of drugs and stabilizes the targeting ability of ferritins [82]. The nano-HAPs also are mineralized on the outer layer of porous scaffold to protect the trafficking and stability of the cell, and balance the biodegradability and antigenicity of cytoplasmic tissues. Then, by changing the ratio of collagen-carboxymethyl cellulose (Col-CMC) from the mineralized carrier, the pore diameter (100–300 μm) of the porous Col-CMC/HAP biomimetic scaffold is suitable for the transport of cellular and cytoplasmic components [83]. For the modification of vaccines, the bio-silica as vaccine vector is accumulated on the surface of Escherichia coli to form a bacillus-shaped mesoporous structure with the average pore size of 8.2 nm. The hierarchical mesoporous structure with high specific surface area of 334.6 m^2^/g contributes to the adsorption of concanavalin A antigens for enhancing immune response [84].

The above examples fully prove that the role of mineralized macropores is mainly to transport large-sized molecules, promote cell adsorption and release, and protect the stability of cells and drugs in the chemical environment in vivo. Using biobased three-dimensional network porous structure and active mineralized particles produces a bone-like support porous structure and mineralized wrapped porous structure. Mineralized stratification structures show great application potential in tissue engineering, vaccine improvement, cell protection, and biomedical therapeutics.

The biomimetic mineralization technique, however, is facing challenges. Although natural porous scaffolds are abundant, their morphologies can hardly be regulated. This is a common limitation of up-bottom strategies. Meanwhile, macropores need to exist in the mineralization process, because mineralized carriers are unable to move fast enough in mesopores and micropores. Mineralized layers can easily block the microporous channels, leading to low dispersion uniformity of mineralized layers in the scaffold. For the same reason, the hierarchical porous structure of living organisms, which has various functions, can be difficultly duplicated by biomimetic mineralization methods [85].

As the above discussion of biomimetic porous processing technology describes, the complex natural pores are replicated from the artificial approach to biomimetic processing technology at micro-, meso-, and macro-scale. With the establishment of the bionic manufacturing method-technology system, the manufacturing scale range of porous materials has been expanded, and its structure-function design theory has been further improved. However, the manufacturing scale of biomimetic processing technology is facing the challenge of unity of accuracy and efficiency. Thus, biomimetic processing technology will be innovated by integrating high-dimensional printing technology and diversified control technologies for improving the accuracy and scale of porous manufacturing structures.

## 3. Structure-Function Relationship of Bionic Porous Structures

### 3.1. Effect of Pore Size on Performance

To design and manufacture the porous, high-performance, multifunctional materials, researchers have carried out a series of studies on the impact of biomimetic porous structures on performance and function. Notably, the scale effect of pore structure is an important factor in regulating the function of porous materials. The scale effect of pore structure also affects the selective adsorption or preferential permeation of active substances due to its different porosity and specific surface area. Then, in biomimetic hierarchical porous materials, micropores, mesopores, and macropores occupy different roles to improve the function and properties of porous materials.

Microporous structures widely exist in the natural tissues of organisms, including roots, leaves, hair, skin, gills, and cell membranes, and play an important role in life activities such as stomatal transpiration, filtration, adsorption, and transfer. In the inspiration from natural microporous structures, a microporous structure is introduced into the design of porous structures to improve the functions of energy absorption, adsorption, catalysis, and support for porous materials. Thus, microporous biomimetic materials can be used in medical treatment, electrode materials, bone scaffolds, catalysts, sewage treatment membranes, electronics, and other aspects. Especially, micropores have the characteristics of selective effect and space restriction effect. Micropores can selectively adsorb or pass through tiny particles of the suitability of size and shape, and trap active particles of a limited space, promoting the selectivity and efficiency of the reaction against porous materials. For example, the limiting properties of microporous structure effectively control hemostasis and improve the wound-healing function of the hydrogels [61]. The microplastic particles and water molecules selectively permeate through the gill-like inclined micropores, which increases the contact timeliness between the microplastic and the pores, and improves the retention rate of microplastics [62].

Mesopores (2–50 nm) have the characteristics of space limitation and molecular recognition, and supplement the accessibility of micropores, which are widely found in bones, horns, bamboo wood, and diatoms. Mesopores with high surface area can be used as carriers for some potential applications of adsorption, separation, catalysis, and bone regeneration. Mesopores also have some special properties of molecular recognition and space limitation. For example, as the active center, the mesoporous structure traps the reactive intermediates in its space, promotes coupling reactions, and achieves efficient catalytic performance [86]. Then, the high specific surface area and rich active sites of the mesoporous structure of the biomimetic probes are used to improve the indocyanine green loading capacity [87]. The mesopores of biomimetic probes improve biosafety, targeting efficiency and photoacoustic imaging stability because of its mesoporous properties of molecular recognition and space limitation.

Macroporous (>50 nm) structures are widely spread in nature such as animal hair, bamboo, wood, blood vessels, veins, and other animal and plant tissues. The main reason why the macropores can provide a wide space and a transport channel is their characteristics of big diameter and large pore volume. Thus, macropores have the functions of heat insulation, permeability, and deformability. For instance, modeled on penguin feathers and polar bear fur, biomimetic macropores of the solar evaporator accelerate ion transport and inhibit salt accumulation to improve efficient evaporation efficiency and salt resistance [88]. The biomimetic macropore (10 μm–1 mm) of the ceramic composite sponge with high porosity (>85%) and high elasticity is mechanically stimulated to control the transport of molecules and cells, as shown in Figure 4 [89].

Furthermore, hierarchical pore structures, consisting of at least two kinds of pores, play a key role in complex life activities. In the process of adsorption or permeation of guest molecules, micropores are selected to pass, mesopores increase the accessibility of passage, and macropores are the main transmission path [90]. Thus, the characteristics of microporous, mesoporous, and macroporous materials are used to improve the mass transfer, catalysis, adsorption, heat insulation, energy dissipation and other properties of biomimetic porous materials, and integrate multi-functionality. For example, compared with mesoporous structure, bionic layered porous carbon has a faster adsorption rate and higher adsorption capacity [91]. Inspired by the biological cell membrane, the hierarchical pore structures (1.4–60 nm), including mesopores and micropores of the carbon membrane, improve the unidirectional ion transport performance and promote forming an asymmetric energy barrier [92].

Then, the synergistic interaction of micropores-mesopores-macropores increases the active sites of guest molecules and promotes the efficient reaction or transport of guest molecules. For example, the meso-macro-porous layered structure of the continuous flow bioreactor stores lipase in the mesopore, while the macropore provides esterification/transesterification reaction space. The synergism of meso-macro pores achieves the significant conversion efficiency, outstanding catalytic activity, and remarkable stability of the bioreactor [93].

### 3.2. Effect of Pore Morphology on Performance

Pore morphology is another one of the significant factors affecting the performance of biomimetic porous materials. The pore morphologies have unique interactions between guest molecules and carriers, resulting in diversities in mechanical properties and functions. Then, the functional integrations of porous materials, including molecular recognition, separation, adsorption, and energy dissipation, are regulated by the pore morphology.

Based on the natural diversity of pore morphology, biomimetic pore morphology has developed many categories such as foam-like [94], honeycomb [9], tubular [11], nanograting [95], jet-like [62], cone shape [96] and truss [88]. For instance, honeycomb pores, inspired by honeycomb, grapefruit peel [9], and diatom shell [4,7], have equivalent wedge effect and large storage space, which are used for the absorption and storage of substances and energy in natural porous material. Similarly, the nanograting biomimetic pore, deriving from the butterfly wing structure, own inherent properties of the periodic order and iridescent structural color, and special functions of radiative cooling, deformation-driven color changing, and interface driving [95]. The tapered nanopore of the biomimetic peptide-gated membrane, comprising a base (510 nm) and a tip (22 nm), can control the transport of substances in vivo and in vitro [96]. Mesh biomimetic pores, resembling an inner structure of penguin feathers, is composed of a three-dimensional truss with connected macropores, accelerates the transport of ions and molecules, and realizes the reflection and absorption of light [88].

### 3.3. Effect of Pore Orientation on Performance

The randomness and complexity of the disordered pore structures has impacted the research of the structural performance relationship, which leads to accumulating key problems of structural reproducibility and stability in the manufacturing process. Compared with the random pore structures, the anisotropic properties of highly ordered pore structure improve the physical properties and mechanical properties of biomimetic materials. For instance, disordered macrocyclic pores hinder the interconnection of cavities and the formation of through-vias, and reduce the effectiveness of the separation function of polymer membranes. By conceiving a cell membrane-like separation structure, the directional stacking method is used to manufacture the ordered pore structure of large ring pores. The aligned-oriented polymer nanomembrane has high selectivity for methanol, and its permeability is twice that of disordered structured membranes [97].

Pore orientation can be regulated by driving forces including single or multiple force fields and intermolecular forces. Then, the orientation of pore structures is controlled by the directional axis to realize the anisotropy of the mechanical properties and optical, electrical, magnetic, and other performance functions of biomimetic porous materials. Biomimetic ordered pores also can be customized by individual functional requirements. For example, referring to the three-dimensional orientation porous structure of the sponge and the honeycomb, the high performances of anisotropic porous materials are obtained in the form of uniaxial or biaxial directional control. Then, by comparing the non-oriented structure, the oriented pore structure is made by controlling of ice-crystal-oriented growth to improve the excellent anisotropic conductivity and sensing performance of biomimetic sponge materials [98]. The oriented pores of cellulose nanocrystalline aerogels are also controlled by the directional assembly of ice crystals and cellulose nanocrystals, and improve the anisotropic piezoelectric performance of biomimetic porous materials [99].

Therefore, the above pore structure factors show that the integrations of the structure and function of biomimetic porous materials can be designed from three aspects. Firstly, the spatial scale effect and hierarchical structure can simultaneously optimize the characteristics of porous materials, such as high specific surface area, multiple active sites, permeability, and multi-response functionality. Next, pore morphology enhances the interaction and molecular recognition performance of guest molecules and pores, which can improve the selectivity and efficiency of functional elements of porous materials. Furthermore, the orientation of the porous structure improves the order and stability of the porous material structure, and increases the anisotropy of the porous material function.

## 4. Design of Porous Meta-Structure

Referring to the natural porous structure-functional relationship, researchers fabricate biomimetic pore structures of high-performance multifunctional materials through the above biomimetic technologies and methods. At present, some research results have verified that the performance and function of artificial structures are beyond that of natural structures. Then, based on the above porous structure-performance relationship theory, researchers customized the high performance and multi-function of pore structure, but the required structure can be unstable in nature. Next, on the basis of structure-performance relationship and manufacturing technology, researchers developed some artificial porous meta-structures involving negative Poisson’s ratio porous structure and DNA origami nanopore structure beyond nature.

In terms of negative Poisson’s ratio porous structure, researchers mainly design some structures, including concave pore structures and origami paper-cut pore structures, from the 2D–3D dimensions to achieve negative Poisson’s ratio effect. Therein, the concave pore structure is regulated by vacuum, force, light, electricity, magnetism, thermal fields, and other external factors. Next, the negative Poisson’s ratio concave pore structure reduces the material density and enhances the mechanical properties of the shear resistance and the electromechanical coupling sensing. For example, the negative Poisson’s ratio concave pore structures of cellulose foam materials are controlled by vacuum gradient and the ionic liquid modification process. Then, the negative Poisson’s ratio concave pore structures improve the conductivity and compressive rebound strain in up to 90% of porous cellulose foam materials [100]. Additionally, based on supercritical foaming, the negative Poisson’s ratio (NPR) value of the concave pore structure is adjusted up to −0.7 by biaxial force fields (Figure 5a). The compression performance of polybutylene succinate(PBS)-NPR foam increases by 600% in radial direction and 270% in axial direction (Figure 5b) [101].

The origami paper-cut pore structure of another negative Poisson’s ratio metamaterial contributes to enhancing mechanical properties, improving scalability, enlarging specific surface area and other characteristics. For the origami paper-cut structure, researchers customized a structural metamaterial with a negative Poisson’s ratio pore structure via 3D printing technology. For instance, using origami paper-cut designs, unique star-shaped perforated morphologies, including three-star, four-star, six-star, and the star-shaped pores of 3D printing metamaterials enhance the tensile strain of 15% due to a negative Poisson’s ratio effect [102]. However, the correlation between negative Poisson’s ratio characteristics and structural deformation is not clear, and the limitations of manufacturing materials and manufacturing technologies obstruct the application of negative Poisson’s ratio porous materials.

Furthermore, the nanostructured DNA origami pore structure is manufactured by the artificial DNA origami technology to break the rules of unstable existence of DNA folding structures. For the problem of a DNA biomimetic template, the high ionic strength is used to stabilize the porous mineralized structure, but hinder the mineralization deposition process. To meet this challenge, researchers create DNA origami technology to form complex nanopore cells such as triangular pores, round pores, and honeycomb pores. For example, DNA origami nanopore structures, from the 1D–3D dimension, are accurately replicated via DNA origami templates to improve the toughness of siliconized porous materials beyond using DNA templates [103]. Additionally, the size of DNA origami nanopore structure can be regulated to overcome the deposition of drugs, and enhance the targeted accuracy and efficient utilization of drugs [104]. Additionally, the DNA origami nanopore structure promoted hydrogen bonding between DNA molecules and friction between DNA chains and T bases at the edge of the pore. For instance, DNA origami-graphene nanopore structures recognize the residence time and ion current of different DNAs to select DNAs [105].

As is mentioned above, the artificially designed porous superstructure breaks the matching limitations of structural functions and the stability limitations of natural structures. The guidance of meta-structure designs can produce supernatural porous materials with excellent mechanical properties, expand the functional integration scale of materials, and improve the efficient utilization of natural structures. However, the diversification of porous meta-structures can be insufficient to meet the requirements of porous structure. Therefore, there are some significant factors that should be paid attention to for the enrichment of porous technology systems and the structure-function relationship, especially pore size, pore morphology, and orientation. Then, the design idea of the bionic-meta structure provides a direction for the adaptability of new high-performance and multifunctional materials and the integrated design of structure and function with high utilization value.

## 5. Summary and Outlook

By learning from natural porous structures and biomimetic pore manufacturing technologies, including the mineralization method and the template method, pore processing technology and its collaborative biomimetic technology are constantly being innovated to create more complex pore structures. Herein, this review mainly discussed in detail the development degree of current biomimetic porous preparation technology from the aspects of manufacturing accuracy, pore structural level, pore scale, and pore order, as follows:(1)The imitation accuracy gradually develops from the macroscopic micron level to the nanometer scale, and the bionic can manufacture complex porous structures more accurately.(2)The biomimetic structure level ranges from single-layer holes to multi-level sub-complex structures such as layered/fractal, enriching the diversity of structures and the integration of structural functions.(3)The pore scale imitation control range is extended, from micron to nano, to realize the refinement of pore structure and the expansion of performance matching.(4)The degree of order of the porous structure is from random to highly ordered, and the reproducibility and functional anisotropy of the structure are realized.

Furthermore, the biomimetic porous structure-function relationship reveals that the scale, morphology, and orientation of pore structures are the important factors affecting the functional integration and performance improvement on porous materials. Biomimetic pore structures have excellent characteristics and various functions such as low-density, high specific surface area; high mechanical strength; functional adsorption transmission; and energy absorption.

In fact, researchers have designed some porous superstructures for breaking the stability limitations of biological structures, crossing the finiteness of biological porous structure utilization, and achieving beyond natural structural characteristics and high performance. In view of the mentioned challenges of biomimetic technology and porous structure design, some constructive suggestions are putting forward as follows:The current bionic structure design and manufacturing still need to develop high-end manufacturing technology to enhance the accuracy and adaptability of bionic methods.The developed high-level substructure can overcome the contradiction between the inherent properties of the porous structure and the functional compatibility, and realizes the high-level integration of functions.The study of the deeper relationship between natural porous structure and func-tion will enhance the application value of biological porous structure.In the future, human beings will expand the types of porous meta-structure de-sign, establish a clear structure-activity theory system of meta-structure, and fur-ther develop porous meta-structure designs.

## Figures and Tables

**Figure 1 biomimetics-08-00140-f001:**
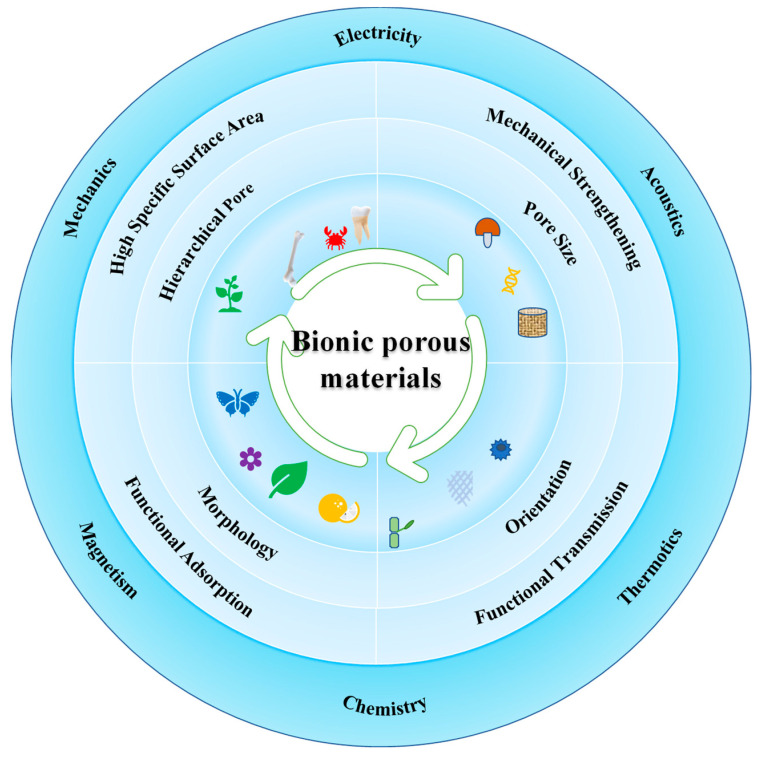
The structure, performance, and application of biomimetics porous materials.

**Figure 2 biomimetics-08-00140-f002:**
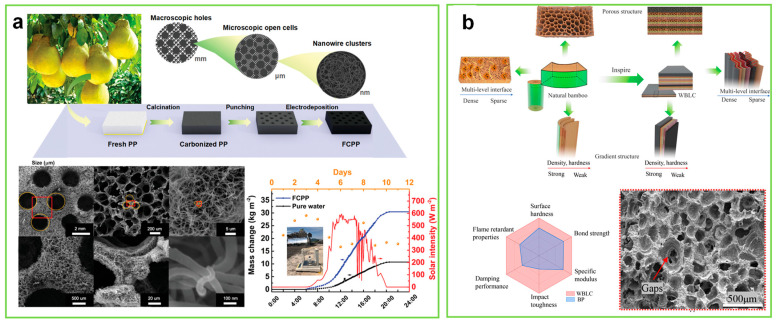
(**A**)The macroscopic processing, pore morphology and application of peel-like pores (Reprinted with permission from ref. [9], Copyright 2020, Wiley-VCH). (**B**) the fabrication and pore morphology of bamboo-like pores (a red frame indicates the surface porous structure of wood veneer/bamboo charcoal plastic/laminated sheet/composite (WBLC)). (Reprinted with permission from ref. [13], Copyright 2022, American Chemical Society).

**Figure 3 biomimetics-08-00140-f003:**
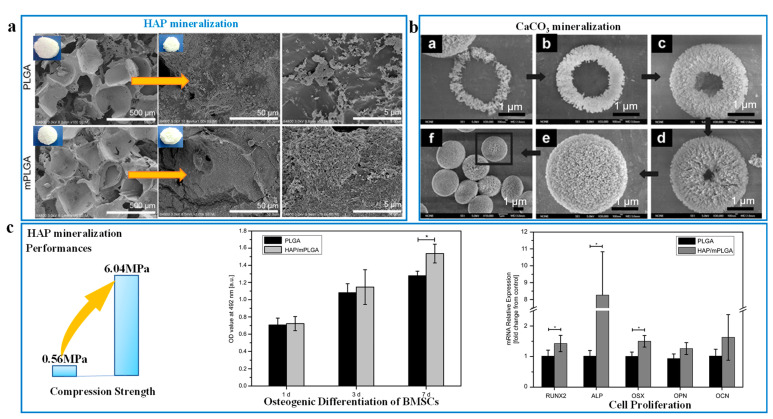
The pore morphology of nanohydroxyapatite particles (nano-HAPs) mineralized on the poly-(lactide-co-glycolide acid) (PLGA)before and after modification (**A**), and the compression strength, osteogeny differentiation and cell proliferation performances (**C**) of the poly-(lactide-co-glycolide acid) porous scaffold before and after mineralization (Asterisks represent data with an error of less than 0.05, which is statistically significant.) (Reproduced with permission from ref. [74], copyright 2018, Wiley-VCH); and the ring-like mineralized growth of the porous CaCO_3_ in figures a–f (**B**) (reproduced with permission from ref. [75], copyright 2008, American Chemical Society).

**Figure 4 biomimetics-08-00140-f004:**
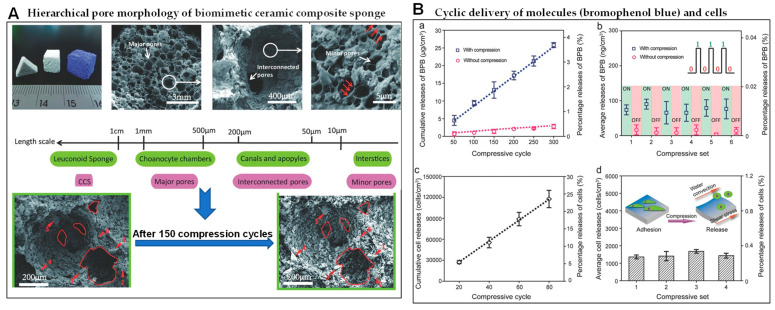
Hierarchical macropores structure, compression and transmission performance of a biomimetic ceramic composite sponge (CCS). (**A**) The sponge-like hierarchical macropores morphology and size of CCS (white as CCS, blue as BPB-loaded CCS), and the microstructure of CCS before and after 150 compression cycles. (**a**) white arrow indicates the position and partial enlarged drawing of major pore, interconnected pore and minor pore; a red arrow indicates the “bridging mechanism” between starch molecules and neighboring ceramic particles; the red outlines describe pore morphology and the numbers indicate large ceramic particles. (**B**) the cyclic delivery of bromophenol blue (BPB) molecules (**a**,**b**) and cells (**c**,**d**) of a biomimetic ceramic composite sponge (Reprinted with permission from ref. [89], Copyright 2017, Wiley-VCH).

**Figure 5 biomimetics-08-00140-f005:**
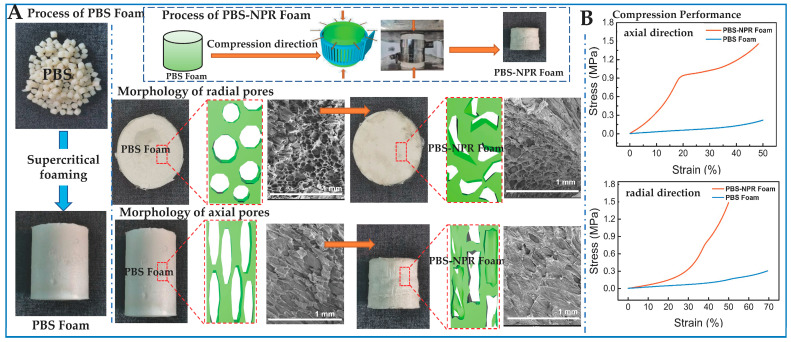
The preparation and pore morphology variation after compression direction of PBS-NPR foam and PBS foam (**A**), and the axial and radial compression tress-train curve of PBS-NPR foam and PBS foam (**B**) (Reprinted with permission from ref. [101], Copyright 2021, American Chemical Society).

**Table 2 biomimetics-08-00140-t002:** Influence of biological template on the structure of biomimetic materials.

Nanomaterials	Biological Template	Pore Size (nm)	SSA (m^2^/g)	Porous Structure	References
SiO_2_	collagen	2.1–4.5	907–1096	hierarchical pores	[39]
	cellulose nanocrystal	-	348–1107	hollow	[40]
	glucose	3.2–3.5	800	mesopores	[41]
	chitosan	3.38	511.77	mesopores	[42]
TiO_2_	cellulose nanocrystal	5.6	13	macropores, mesopores	[43]
	cellulose nanofiber	6.8–7.5	53–67	hierarchical pores	[44]
	chitosan	9	4	mesopores	[45]
	agarose	4.7	121	porous network	[46]

## Data Availability

Not applicable.

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
