# Peer review of "Design, Manufacturing and Functions of Pore-Structured Materials: From Biomimetics to Artificial"

_biomimetics, 2023, doi:10.3390/biomimetics8020140_

Round 1
Reviewer 1 Report
The subject is timely and of great interest. The authors presented a review of design, manufacturing and functions of pore-structured materials. The manuscript was well written and the discussion is of in depth. The reviewer believes this is a paper of high quality and it is suggested that the article be accepted after the following minor comments are taken care of. I will be happy to review the revised manuscript again.
1. Bamboo is a blueprint porous material that exhibits high strength and toughness. The authors need to talk about bamboo’s strengthening and toughening mechanisms and how to make bamboo-like materials with reference to the following paper.
In Situ Observation of Fracture Behavior of Bamboo Culm, JOM, 73 (2022) 1705–1713
Nanoscale structural and mechanical characterization of the cell wall of bamboo fibers, Materials Science and Engineering C, 29 (2009) 1375-1379
Nanoscale structural and mechanical characterization of bamboo-like polymer/silicon nanocomposite films, Nanotechnology, 16 (2005) 1746
22. Biomaterials have porous architecture that can also serves as templates to derive nanomaterials which may find application in many fields. The authors need to present such aspects with reference to the following papers.
A generic bamboo-based carbothermal method for preparing carbide (SiC, B4C, TiC, TaC, NbC, TixNb1− xC, and TaxNb1− xC) nanowires, Journal of Materials Chemistry 21 (2011) 9095-9102
TaC nanowire/activated carbon microfiber hybrid structures from bamboo fibers, Advanced Energy Materials, 1 (2011) 534-539
Biotemplating fabrication, mechanical and electrical characterizations of NbC nanowire arrays from the bamboo substrate, Journal of Alloys and Compounds 560 (2013) 142-146
33. Seashells also are a type of biomaterial that show great inspiration for biomimetics. Thermally treated conch shell and nacre are porous materials. The authors need to extend the discussion on such aspect with reference to the following paper.
Multiscale hierarchical assembly strategy and mechanical prowess in conch shells (Busycon carica), Journal of structural biology, 184 (2013) 409-416
Structural and mechanical characterization of thermally treated Conch shells, JOM. 67 (2015) 720-725
Dynamic Self-strengthening of A Bio-nanostructured Armor - Conch Shell, Materials Science and Engineering C, 103 (2019) 109820.
Nanoscale structural and mechanical characterization of natural nanocomposites: seashells, JOM, 59 (2007) 71-74
Reviewer 2 Report
This review work aims to summarize the existing methods for preparing biomimetic porous structures, covering various applications such as the biomedical field, energy absorption, separation, bone work, etc. The author team concludes the methods, processing techniques, and key factors, and also introduces meta-structures to achieve some superstructure that is hard to achieve in nature.
Overall, this work has a relatively clear outline and layout. Major comments are listed below:
1. Some sections are suggested to be modified or combined with others. For example, Section 2.1, the mineralization method, is better described as a method to modify the porous structure instead of fabricating it. They still need a base template as the porous structure/scaffold, which is then mineralized. Thus, it's better to highlight it as a modification way instead of a fabrication method and move it to the latter section.
2. The next major section, template method, is better characterized by all the methods in this section as random template (ice, freezing way), naturally-derived template (bionic template directly from the bio-template), and customized-design template (inspired by the bionic structure; mimic but not fully replicated). Then it will be clear about the difference of those templates, and their advantages, or disadvantages.
3. There is a lack of review on different processing technologies to achieve "top-down" structures, such as electrospinning, phase-inversion, freeze-drying, etc.
That's the core concern the reviewer has from the whole manuscript.
Besides these main points, other concerns/questions arise from the reading of the paper. Please address them in the revised work, Please address them in the revised work, meeting the requirement for publication in Biomimetics
· Abbreviations: Please use the full name when you introduce the abbreviations for the first time.
· Figure 2: 1) missing information for Figure 2a; 2) Reorganize Figure 2, and use the bold frame to separate b and c. 3) in Line 114, Figure 2C you adapted here covered mechanical performance AND cell proliferation performance. Move it to the end.
· Line 114: CaCO3 seems like another modification, similar as Nano-HAP. So “then” is not properly used here. Besides this connection, what’s the performance of modified groups, any property changes or improvements?
· Section 2.1: Not a fabrication method, more like a modification method.
· Line 174: How do you define the term “Anisotropic” here? How does it come?
· Line 177: Besides improving the mechanical performance, the porous structure can also be used to confer flexibility and tailor the mechanical performance. A polymeric porous foam, which was fabricated via freeze-drying, can mimic the native spongiosa layer of the aortic valve, in order to replicate the load-bearing role and tune the mechanical properties (Sun et al, Biomaterials 288, 121756, https://doi.org/10.1016/j.biomaterials.2022.121756) Thus, it’s better to use “adjust” or “tune” to replace “improve” under this circumstance.
· It’s impressive to see the summary of the problems from the application of biomimetic materials, in line 232. Good job.
· Line 249: How do they process/fabricate their porous material (Your reference 13)? You should focus on processing details, instead of property/performance description in this section. Please specify their methods for the layered porous structure
· Figure 3b: needs a high-resolution image
· Section 4: It’s better to adapt figures to illustrate negative Poisson’s ratio porous structures (concave & origami paper-cut pore structures) and explain how they work, as what you did in Figure 3
· Line 472: “There’re some significant factors, what are they”. Please specify them.
